# Conservation through Biocultural Heritage—Examples from Sub-Saharan Africa

**Anneli Ekblom [1,2,\*]**, **Anna Shoemaker [1]**, **Lindsey Gillson [3]**, **Paul Lane [1,4,5]** and **Karl-Johan Lindholm [1]**

1. Department of Archaeology and Ancient History, African and Comparative Archaeology, Uppsala University, Box 626, SE-751 26 Uppsala, Sweden; anna.shoemaker@arkeologi.uu.se (A.S.); paul.lane@arkeologi.uu.se (P.L.); karl-johan.lindholm@arkeologi.uu.se (K.-J.L.)
2. Natural Resources and Sustainable Development, Department of Earth Sciences, Uppsala University, Villavägen 16, 75236 Uppsala, Sweden
3. Plant Conservation Unit, Botany Department, University of Cape Town, Private Bag X3, Rondebosch 7701, South Africa; lindsey.gillson@uct.ac.za
4. Department of Archaeology, University of Cambridge, Downing Street, Cambridge CB2 3DZ, UK
5. School of Geography, Archaeology and Environmental Studies, University of the Witwatersrand, Johannesburg 2000, South Africa
\* Correspondence: anneli.ekblom@arkeologi.uu.se

**Abstract:** In this paper, we review the potential of biocultural heritage in biodiversity protection and agricultural innovation in sub-Saharan Africa. We begin by defining the concept of biocultural heritage into four interlinked elements that are revealed through integrated landscape analysis. This concerns the transdisciplinary methods whereby biocultural heritage must be explored, and here we emphasise that reconstructing landscape histories and documenting local heritage values needs to be an integral part of the process. Ecosystem memories relate to the structuring of landscape heterogeneity through such activities as agroforestry and fire management. The positive linkages between living practices, biodiversity and soil nutrients examined here are demonstrative of the concept of ecosystem memories. Landscape memories refer to built or enhanced landscapes linked to specific land-use systems and property rights. Place memories signify practices of protection or use related to a specific place. Customary protection of burial sites and/or abandoned settlements, for example, is a common occurrence across Africa with beneficial outcomes for biodiversity and forest protection. Finally, we discuss stewardship and change. Building on local traditions, inclusivity and equity are essential to promoting the continuation and innovation of practices crucial for local sustainability and biodiversity protection, and also offer new avenues for collaboration in landscape management and conservation.

**Keywords:** biocultural heritage; sub-Saharan Africa; traditional ecological knowledge; hotspots; sacred forests; conservation

---

## 1. Introduction

Globally, a high proportion of biodiversity resides outside of protected areas. Incentives for biodiversity protection, therefore, must be built and fostered amongst diverse stakeholders, in areas where biodiversity and communities co-exist [1,2]. In keeping with this principle, biocultural heritage is an emerging concept drawing on local knowledge, land-use practices and heritage values to define sustainability and resilience from the perspective of local inhabitants [3–10]. The concept is particularly relevant in African contexts, as many landscapes can be defined as *continuing* cultural landscapes following the International Union for Conservation of Nature (IUCN)'s definition of its category V

landscapes as those "where the interaction of people and nature over time has produced an area of distinct character with significant ecological, biological, cultural and scenic value" [11]. In rural areas of Africa, the most common forms of agriculture entail low-intensity land-use practices, often based on various customary systems of access and ownership rights. Globally, there has been increasing realisation that the discontinuation of small scale, low intensity agricultural practices contributes to the recent reductions in biodiversity [12–16]. However, in many African settings environmental debates are still centred on the assumption that local practices of fire management, cultivation and/or grazing cause degradation (for summaries and critiques of such arguments, see, e.g., [17–24]). As will be exemplified here, local low intensity and customary practices may hold the key to strengthening, adapting and re-innovating forms of land-use that accommodate biodiversity and cultural heritage and promote adaptive management and resilience [25]. Building on and reinvigorating such local practices is important given that the effects of climate change are accelerating and climatic insecurity and its effects on food production and security are increasingly pertinent issues [26–28]. At the same time, ongoing competition for land from industrial agriculture, biofuel production, carbon off-setting projects and conservation initiatives make local communities increasingly vulnerable to both climate change and socio-economic transformations that are detrimental to particular livelihood traditions [29–33].

## 2. Background

As an emerging field, biocultural heritage has been explored from different disciplinary perspectives ranging from those focused on socio-cultural practices explored using ethnographic methods, to those rooted in understanding and modelling biological systems on a grand scale [5,6,9,34,35]. The origins of the concept can be traced back to the emerging interest in community-based resource management and traditional ecological knowledge in the 1980s [36–39], the adoption of the Convention on Biological Diversity (CBD), and the aftermath of the 2003 IUCN World Parks Conference in Durban, South Africa [4]. The United Nations Educational, Scientific and Cultural Organization (UNESCO) uses the term 'biological cultural heritage' to refer to ecosystems (including habitats and species) originating or developing from human practices [40]. More broadly, biocultural heritage is considered to encompass the natural–cultural components of human–environment interactions including knowledge, practices and innovation. Practices related to biocultural heritage are also closely linked to the construction and confirmation of identities and social cohesion [25,41–45]. Biocultural heritage has been key in developing both local advocacy groups and legal frameworks focused on the protection and ownership of landscapes and resources by and for local communities [4,46–48]. The concept has also been incorporated into conservation biology and broadened to include deep-time landscape history [7,8,49,50].

While there are now several different conceptualisations of biocultural heritage (see, for instance, [4,5]), we draw on the framework developed by Lindholm and Ekblom [50] in defining biocultural heritage as consisting of four interactive elements, each operating at interlinked temporal and spatial scales, that can only be understood through integrated landscape analysis. *Ecosystem memories* (Figure 1) can be defined as practices and outcomes operating on larger or deep-time scales, where agricultural, grazing and/or fire management activities have reshaped landscapes with long lasting effects on both biological and landscape structures. *Landscape memories* represent smaller scale materialised human practices and ways of organising landscapes and their outcomes. These include changes in soils, geological formations, flora and fauna but also archaeological sites, built environments and living land-use practices. Local heritage practices and narratives are often interlinked with such land-use activities and play a vital part in maintaining them. *Place memories* are also defined by local narratives, place names and signs of earlier or continuing practices whose significance is under constant debate and re-negotiation both locally and with external actors. These memory elements will be exemplified in more detail below. The fourth element, *stewardship and change*, concerns the conceptualisation and transfer of knowledge pertaining to landscape

management, collaborative innovation and self-determination. The fusion of biodiversity goals with social and economic goals founded on self-determination is essential for establishing ecologically sound and equitable landscape management practices, as we explain below. To identify and explore how and why these four elements intersect requires *integrated landscape analysis*. These inclusive methodological and conceptual approaches to knowledge and landscape management are essential to both documenting and researching biocultural heritage and applying the insights generated to future stewardship and adaptive practices.

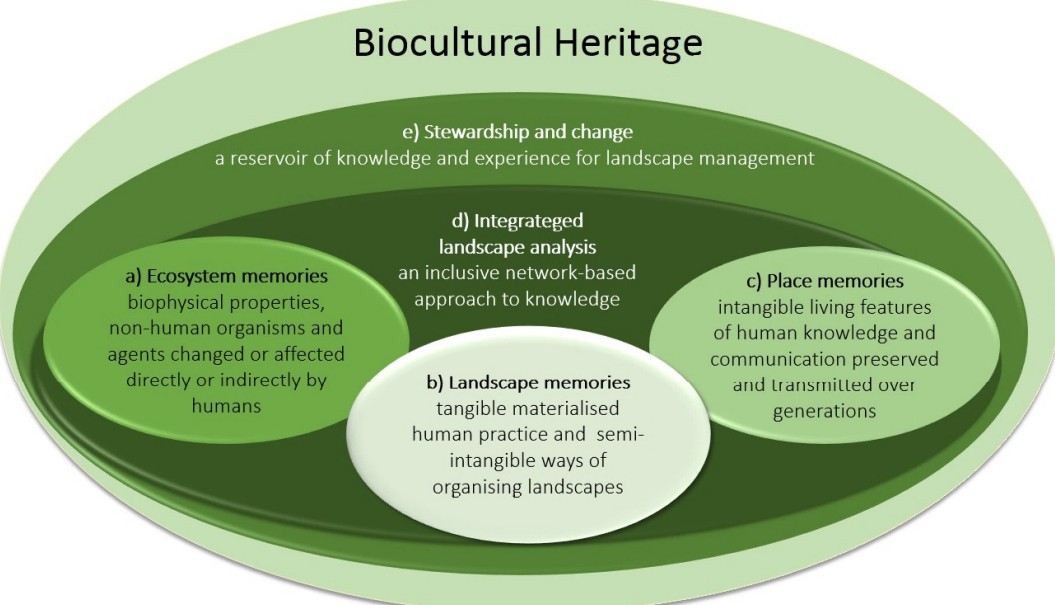

**Figure 1.** The four 'elements' of biocultural heritage (**a–d**) as defined by Lindholm and Ekblom [50].

More specifically, integrated landscape analysis is an interdisciplinary toolbox allowing us to trace elements of biocultural heritage and their internal relationships over time, incorporating contemporary botanical surveys, pollen analysis, archaeology, geographical information systems, cartography, historical research and interviews, and participant observation [50]. Studies spanning over both millennia long and shorter timescales have been critical in terms of understanding the biophysical and social ecological aspects of biocultural heritage. Such studies have also been important for re-evaluating degradation narratives that risk impeding effective biodiversity conservation and landscape management practices. A good early example of this is provided by the work of the People, Land and Time in Africa (PLATINA) research group at Stockholm University on deconstructing environmental narratives concerning the origins and drivers of severe soil erosion in Kondoa District, central Tanzania, that had informed colonial and post-colonial interventions for decades (see [51–53] for summaries, and additional references). A deeper understanding of how humans shape landscapes is an essential component in any plan for a sustainable future [7,54–62].

Below we review examples of land-use practices in sub-Saharan Africa, structured around the five interactive elements of biocultural heritage. We combine biological inventories and/or assessments by local residents on the ecological effects of land-use practices, with condensed summaries of archaeology, vegetation history, and interviews with local practitioners. We also complement these with our own and our students' field studies. In none of the cases presented below can the elements of bicultural heritage be understood from one single discipline nor from one single vantage point, whether that is the perspective of a local herder or farmer, researcher, conservationist, development worker or government official.

## 3. Biocultural Heritage in Space and Time

### 3.1. Ecosystem-Scale Memories

The concept of 'ecosystem memories' comes closest to UNESCO's [40] definition of biocultural heritage as ecosystems developing from human practices. Meadow pastures and wood pastures in Africa can be seen as continuing cultural landscapes, found in savannas and woodlands that are grazing- and fire-dependent [63–66]. Agroforest- and fire- managed landscapes carry structural and species level memories in terms of biological diversity. Interdisciplinary studies are still too few to allow us to assess the performance of these landscapes in terms of biodiversity, but broader scale studies on fire ecology in Africa have shown that landscapes that are fire-managed regularly tend to have both lower intensity and cooler fires that occur earlier in the dry season and which are more beneficial to sustaining biodiversity [65–67]. Fire also plays a crucial part in local landscape management in the semi-dry regions of southern and eastern Africa. In the savanna shrubland of the Chyulu area, Kenya, people burn to improve either hunting or pasture by, in the case of the latter, removing unpalatable grass and ticks. Grazing and dead wood collection are also used to create fire breaks [68]. Such systems are practiced in several regions of southern Africa, but the Chitemene system in Zambia has become particularly renowned for promoting pasture and fertilisation of farms, while also protecting individual trees, and thereby creating fire breaks [69]. In savannas, such mixed fire regimes and patch mosaic burning results in a heterogenous landscape structure comprising a range of post-fire ages, favourable to biodiversity, but managed fires also prevent damaging late season hot fires and uncontrollable wildfires that homogenise landscapes and eliminate fire-sensitive species [65,66,70].

In forested regions, agroforest landscapes create parkland and mosaic landscapes that are structurally diverse and high in agro-biodiversity [71]. The parkland mosaic landscape has a continuity over millennia. Previously, linkages were made between the extent of savannas and parklands and degradation from farming and fire management going back ca. 4000 years. However, palaeoecological studies now suggest climate dynamics have been more important in shaping the distribution of savannas in West Africa over the long term [72,73]. However, humans have also contributed to shaping mosaic landscapes, and fires are an important tool for maintaining landscape structure. In the Koulikoro district of southern Mali, fire is used to create landscape mosaics that increase micro- and edge-habitats, which are favourable for biodiversity. Here, fires and clearings create mosaic landscapes of semi-open areas, fields, fallows, and old growth forests [74,75]. Similarly, in the Kissidougou savanna region in Guinea, West Africa, forest patches and boundaries are continuously protected by households using methods that include mounding (to encourage plant growth), mulching, tilling, planting of crops beneath the trees and protection of tree species [76]. With sufficient fallow periods, such parkland management has positive impacts on tree biodiversity, as has been shown by studies in southwestern Burkina Faso [77]. In East Africa, diverse parkland landscapes are created through a variety of off- and on-farm management and forest protection, resulting in a diversity of trees [78,79]. Apart from shaping landscape structure and biology, fire management and shifting agriculture leave memories in terms of soil nutrients. The combination of burning and mulching of soils leads to the formation of black earths [76,80,81]. These black earths are conducive to both agricultural and biological diversity, as they are higher in organic carbon, pH, and plant-available nutrients than other local soils, and are also less prone to nutrient leaching or acidification [81].

Another similar example of long-term 'soil memory' important for ecosystems and agrobiodiversity comes from the East African savannas. Historic occupations have resulted in the enrichment of soils from dung and the formation of grassy glades [82–86]. In Kenya, pastoralists recognise these glades as marking former settlement sites and value them for the nutrient-rich grasses, especially *Cynodon* spp., that recolonize these former 'human' spaces [87,88]. Interviews and archaeological data presented by Shoemaker [89] covering the last ca. 150 years show how these places have been resettled over generations and are preferred sites for settlement. These 'anthropogenic' glades, described by Veblen [90,91] as biological 'hotspots', produce good pasture grasses and are

functionally important for a range of taxa, including wild megaherbivores [91], but also birds [92,93], geckos, and arthropods (e.g., [94], see also [82,85]). As pastoral communities have been highly mobile over time and soil nutrient compositions can last for millennia [87,88,95–97], the total surface area of such glades is important for overall ecosystem functioning in East African savannas.

### 3.2. Landscape Memories

Landscape memories can be understood as forms of materialised human practice, such as built environments and archaeological sites, including settlement systems and land-use systems linked to user and property rights—what Widgren [98,99] has called 'landesque capital'. Across the continent, there are many examples of irrigation or terracing system landscapes that are relict (see for instance Engaruka in Tanzania [100–102], Nyanga in Zimbabwe [103] and Mpumulanga in South Africa [104]).

In addition to these discontinued terrace systems, there are also landscapes where precolonial irrigation practices are extant (see, for instance, the Mbulu Highlands, Tanzania [105,106]; and the Cheranagni escarpment, Kenya [107]). We will here expand on one example, the furrow irrigation found on the southern slopes of Mount Kilimanjaro, termed *mfongo* by Chagga-speaking people (Figure 2). The earliest firmly dated irrigation features on Kilimanjaro were built in the 18th century, though oral traditions, historical references, and linguistic evidence indicate that irrigation schemes were present on the mountain by the 17th century at the latest [108,109]. This system, built on customary land tenure, supports multi-layered agroforest gardens high in biodiversity (500 species of which 400 are non-cultivated) [110,111]. Despite their productivity and longevity, government policies have long been remiss in terms of promoting and maintaining *mfongo* practices [108,112,113]. Sunday [114] has explored the continuity and legacy of the *mfongo* through interviews with 200 households. The practice of making and maintaining water channels is less common today than it was in the remembered past. When asked why *mfongo* practices are discontinued, local residents replied that increasing droughts and lessened water runoff from Mount Kilimanjaro were the primary causes, although state policies were also mentioned for the wider contextual setting (see also [113]). However, local residents still value the *mfongo* system for ensuring water access and for maintaining crop yields and diversity. The water channels are also embedded in local heritage and identity. In the interviews conducted by Sunday [114] (p. 48), one elderly woman expressed her worry that this knowledge was now being lost:

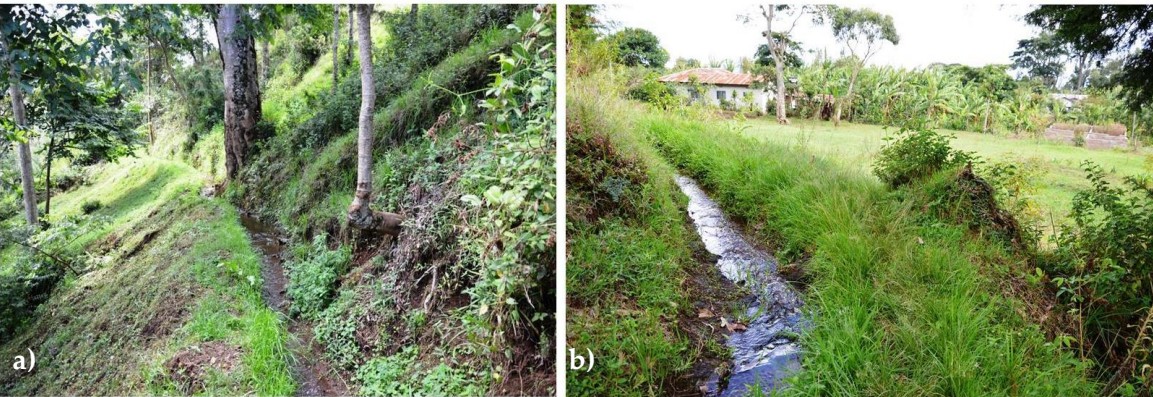

**Figure 2.** (**a**) a traditional water furrow (*mfongo*) cutting across a hillside on Mount Kilimanjaro; (**b**) a water furrow leading up to a Chagga home garden (photos published by Sunday [114] and reproduced with his permission @ Sabbath Sunday).

"Times have changed very fast! Our children who go to school, when they live in cities for long and God blesses them with children: those children will not produce a generation that will revive culture practices of *mfongo* and rituals".

For that woman, and also many other respondents interviewed by Sunday, the making of water channels is more than an agricultural practice, it is also a 'spiritual obligation' and a living knowledge that must be actively maintained.

### 3.3. Place-Based Memories

Place-based memories refer to intangible living features of human knowledge and communication: know-how, place names, orature, arts, ideas and culture, and also biological heritage. A good example is the practice of the protection of 'sacred areas', often ancient and/or community burial areas, or old settlement sites that are surrounded by very strong rules of community protection, some of which have become havens for old growth forests. Though together they may constitute a landscape memory, the foundation of protection is very much placed-based. Sacred sites/forests occur across the African continent, but their potential in terms of biodiversity protection has only been recognised in the last few decades [115–122]. A study in northern Tanzania (the North Pare mountains) located 290 sacred sites. Though small in size, together they covered a total area of 370 ha [123]. Similarly, on Zanzibar there are a great number of sacred areas that are important as reservoirs for the endemic Zanzibar-Inhambane forest phytochoria, which are known to be high in species diversity [124]. Another well-known example is the World Heritage sacred Mijkenda Kaya forests in Kenya. This heritage landscape consists of 11 separated forest islands (growing on abandoned occupation settlements) containing as many as 307 species that are listed as endangered [125,126] (Figure 3). These are sanctuaries for forest-adapted species, both plants and animals, and have been estimated to comprise 4.2–5.6% of the entire Zanzibar-Inhambane forests mosaic [127]. These sacred areas and the knowledge systems surrounding the activities that brought them into being are as key to the transmission of traditional practices as to the innovation of new ones [128,129].

In the Muzarabani Communal Lands of Zimbabwe (Zambezi valley), forest loss has been shown to be dramatically less in areas considered sacred, or under protection of traditional custodians [130]. Similar observations have also been made in Mozambique in the Licuati forest in the south [131] and the Chôa Highlands in Manica Province. In Chôa, sacred areas have greater species diversity, more complex forest structure, and higher incidence of fire-sensitive species [132]. Palaeoecological techniques have been used in southern Mozambique [133] to study the long-term history of littoral forests, suggesting the existence of mosaic landscapes for 1400 years. Though more studies are needed, it has been suggested that existing forest patches should possibly be re-assessed as having been actively protected through long-term (i.e., over centennial scales) management [133]. Though not supported by palaeoecological data, long-term protection of forest patches has also been suggested for East Africa [134]. In Madagascar, there are also positive examples of customary protection of sacred forest. One example is Ankodida, where protection builds on customary rules surrounding resource use and local custodianship [135]. Such culturally protected small forest islands have been shown to be essential for maintaining ecosystem services [136].

Sacred areas are also common in West Africa. In Benin, 2940 sacred areas have been documented ranging in size from 0.1 ha to 1600 ha, covering a total of 18,360 ha [137,138]. Apart from being heritage places, these areas are highly important as biological refugia and function as seed-banks and genetic reservoirs [139]. In numerous regions of Benin, sacred areas can exhibit higher tree species diversity than state-established conservation spaces [140]. Estimates in Ghana suggest there are 2000–3200 such sacred sites [141]. Studies within sacred groves on the Accra Plains found that the biomass and diversity of small mammals often exceed that of surrounding biomes [141]; similar correlations have also been made more recently regarding butterfly populations [142]. In the Loma area in Liberia, dominated by the Upper Guinea forests, sacred areas are often the sites of old towns or graves and the long history and mobility of people and settlements has created a dense network of such sacred areas [143]. Socially proscribed systems of protection surrounding sacred areas allow for the maturation of old growth trees. At the same time, economically important shade-tolerant tree crops (e.g., Kola, cocoa and coffee) can be planted amongst the trees, thriving in the nutrient-dense

soils that often result from anthropogenic inputs associated with settlements, such as charcoal, animal dung and food refuse. A comparison between the sacred areas and unmanaged fallow areas shows that vegetation in sacred areas is more heterogeneous in the basal layers and higher in the upper layers of the canopy [143].

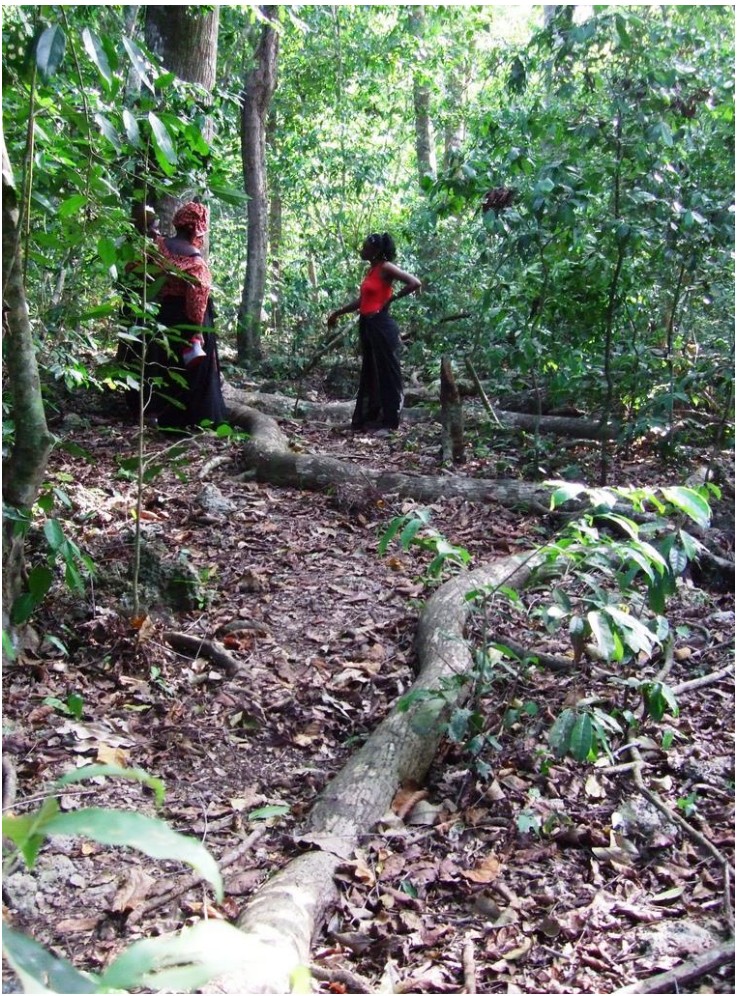

**Figure 3.** Mijkenda Kaya forests in Kenya (picture from Wikimedia commons, photo by Victor Ochieng @Victor Ochieng).

## 4. Discussion: Stewardship and Change

The memories exemplified above act on different spatial and temporal scales shaping the structure of the ecosystem or landscape. However, the practices that shape them are usually place bound. As physically discernible memories (shown through landforms, vegetation and archaeological and/or heritage sites), they inform practices in the present. Biocultural heritage is closely connected with identity, social cohesion and practice but also social and political negotiation locally [25,42,43]. Heritage places, in particular, are often the arena for community meetings where this negotiation is played out. Such events frequently feature the transmission of old and new knowledge [127–129]. Biocultural heritage may then be considered to build on local practices and initiatives, and is therefore key in promoting stewardship, innovation and change [4–6,9,10].

As exemplified in the case of the *mfongo* system, but also more broadly in the case of fire management [20], state and development policies have tended to work against local practices. After nearly a century of state fire suppression polices across the continent, there is now an expanding body of research calling for better integration of fire management and conservation goals and hence closer

collaboration between conservation workers and local farmers and pastoralists [67,74,144]. For instance, in the Chyulu Hills area of Kenya reviewed above, local residents lament the fact that they are not allowed to pick dead wood or instigate controlled burns around park boundaries, which increases the likelihood of uncontrolled fires within the park, and both local residents and conservationists are stressing the need for collaboration [68]. The Caprivi Integrated Fire Management Programme in Namibia is a positive example here: through collaboration, fire management policies have changed from a centralised fire suppression policy to community-managed fire areas where early season burns are used as a fire management tool by local communities [145].

There is also now a growing awareness of the potential for customary practices and heritage sites to promote forest conservation [119,121,127]. A positive example is the Ankodida forest in Madagascar where the World Wild Fund for Nature (WWF) worked with local residents to incorporate sacred groves into the newly forming protected area network. As communities were concerned that erosion of traditional values was leading to threats to the groves, the formal conservation status was seen as reinforcing cultural taboos promoting the protection of the forest [135]. The Mijkenda Kaya forests in Kenya, discussed above, provide another positive example of stewardship based on local practices of protection and heritage values, where local communities and authorities have collaborated to proclaim the area as a World Heritage Site [127], though as discussed below, there are also problems here with inclusivity and access.

One of the biggest challenges since the emergence of community-based resource management (CBRM) has remained the definition of community itself [36,146,147]. Individual community representatives may become a shorthand for 'community', which risks entrenching local power structures that are less than equitable, in particular when it comes to issues related to heritage, often in control of male elders and/or particular lineages. Continuing with the case of the Mijkenda forests, male elders lament the loss of respect for traditions as well as the loss of traditions themselves, processes they believe will ultimately threaten the protection of the sacred forests [127,128]. Meanwhile, based on interviews carried out by Groh [129], youths and women report experiencing issues surrounding access to sacred forest areas, and also the transfer of knowledge and innovations occurring within them during community meetings and ceremonies. For both community leaders and for other local people not so clearly in positions of power, this feeling of exclusion is then a double loss, as crucial knowledge regarding agrobiodiversity and resource use is not being transferred to younger generations (nor women) and the traditions and values associated with forest protection are thus not prioritised by younger community members. Biocultural heritage, while featuring existing (and customary) local practices of land-use and heritage management must therefore also build on principles of inclusiveness and transparency, otherwise it will simply not be accepted by the wider community.

When integrated with principles of equity, justice and representation, while also building on local practices and innovation, biocultural heritage has the promise of combining the goals and aspirations of local residents with the national and international goals of sustainability and biodiversity protection [148]. Concrete collaborations between local communities and conservation in biocultural heritage are still too few and far between, but as argued here, there are ample opportunities to learn from existing local practices of biodiversity stewardship but also from local processes of agrobiodiversity innovation and change. To conclude, in this paper we have discussed a series of cases illustrating how a conceptual framework of biocultural heritage allows for new approaches to heritage, nature conservation, landscape planning and development goals. Biocultural heritage assets, we argue, provide the means to negotiate management goals in these areas, and in certain cases, also to combine them.

**Author Contributions:** Main author: A.E. Author contributions are as follows: Conceptualization: A.E. and K.-J.L.; Writing—Original Draft Preparation: A.E., A.S.; L.G.; P.L.; K.-J.L.; Writing—Review and Editing: P.L., A.S. and L.G.; Visualization: A.E. and K.-J.L.

**Funding:** The paper has been partly produced within the Adaptation & Resilience to Climate Change (ARCC) in Eastern Africa project funded by the Swedish Research Council (Vetenskapsrådet), Formas and the Swedish

International Development and Cooperation Agency (SIDA), grant number 2016-06355, awarded to Paul Lane and Anneli Ekblom. Anna Shoemaker's work was supported as part of the European Commission Marie Skłodowska-Curie Initial Training Network titled "Resilience in East African Landscapes (REAL)" (FP7-PEOPLE-2013-ITN project number 606879, awarded to Paul Lane).

**Acknowledgments:** This article is a contribution to the Integrated History and Future of People on Earth (IHOPE) initiative (http://ihopenet.org/), and to the Pages LandCover6K and LandUse6K Africa working group activities.

**Conflicts of Interest:** The authors declare no conflict of interest.

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
