# Peer review of "Conservation through Biocultural Heritage—Examples from Sub-Saharan Africa"

_land, doi:10.3390/land8010005_

Round 1

Reviewer 1 Report

Overall this is a really important paper and should be published. I have a few comments for improvement:

- line 40, 43: what do you mean by 'local practitioners'? I assume you mean local communities? But it could also mean government or NGO practitioners, best be clear.

- line 44: I am not sure it is correct to say that the concept is particularly 'useful' in African contexts  since it has been more widely researched and applied in Latin America - perhaps replace with 'very relevant' 

- line 55: after 'cultural heritage' I would add 'and promote adaptive management and resilience' - could refer to Berkes (Ref 40). 

- line 72: Best add a reference here - Swiderska (2006). Banishing the Biopirates: A new approach to protection traditional knowledge. Gatekeepers 129. IIED, London

- Line 95: I find it a bit odd to include methods and approaches as part of a concept which essentially reflects the holistic indigenous worldview - or at least that is how the indigenous NGO ANDES (Peru), IIED and partners defined biocultural heritage in 2005 (see Gatekeepers 129)

- Line 101 - probably don't need a separate section here

- line 107-109 - ideally add a sentence to explain why/how such studies have been important for re-evaluating degradation narratives

- 117: how are local practitioners different from the others? 

-121: 29 is not a UNESCO reference, should it be 39?

-155, what is 'cation'?

-162: add 'that' after 'places' 

- 181: do you mean figure 2? 

- 213: could briefly explain what is meant by 'red-listed'

- 256 - delete 'relevant'?

- 270: replace 'about' with 'that'

- 271- change 'statues' to 'status'?

-284 - sentence could be clearer

Author Response

Overall this is a really important paper and should be published. I have a few comments for improvement:

- line 40, 43: what do you mean by 'local practitioners'? I assume you mean local communities? But it could also mean government or NGO practitioners, best be clear.

RESPONSE: NOW REFORMULATED

- line 44: I am not sure it is correct to say that the concept is particularly 'useful' in African contexts  since it has been more widely researched and applied in Latin America - perhaps replace with 'very relevant' 

RESPONSE: NOW REFORMULATED

- line 55: after 'cultural heritage' I would add 'and promote adaptive management and resilience' - could refer to Berkes (Ref 40). 

RESPONSE: NOW REFORMULATED AND REF MOVED

- line 72: Best add a reference here - Swiderska (2006). Banishing the Biopirates: A new approach to protection traditional knowledge. Gatekeepers 129. IIED, London

RESPONSE: NOW ADDED

- Line 95: I find it a bit odd to include methods and approaches as part of a concept which essentially reflects the holistic indigenous worldview - or at least that is how the indigenous NGO ANDES (Peru), IIED and partners defined biocultural heritage in 2005 (see Gatekeepers 129)

RESPONSE: WE HAVE NOW REWORDED THIS CONSIDERING THE COMMENT TO SAY FOUR ELEMENTS

- Line 101 - probably don't need a separate section here

RESPONSE: SECTION NOW DELETED

- line 107-109 - ideally add a sentence to explain why/how such studies have been important for re-evaluating degradation narratives

RESPONSE: NOW ADDED

- 117: how are local practitioners different from the others? 

RESPONSE: NOW REWORDED

-121: 29 is not a UNESCO reference, should it be 39?

RESPONSE: NOW CHANGED

-155, what is 'cation'?

RESPONSE: NOW EXPLAINED

-162: add 'that' after 'places' 

RESPONSE: NOW ADDED

- 181: do you mean figure 2? 

RESPONSE: NOW CHANGED

- 213: could briefly explain what is meant by 'red-listed'

RESPONSE: NOW REPHRASED TO “that are listed as endangered”

- 256 - delete 'relevant'?

RESPONSE: DONE

- 270: replace 'about' with 'that'

RESPONSE: DONE

- 271- change 'statues' to 'status'?

RESPONSE: DONE

-284 - sentence could be clearer

RESPONSE: NOW REPHRASED

Reviewer 2 Report

I enjoyed reading this manuscript. It is an interesting paper that considers the important inter-relationships between humans and the environment.  It acknowledge the strong cultural connections with landscapes that many societies have and recognises the importance of local scale and historical agricultural and land use practices in shaping the landscape.  Consideration to 5 key elements that make up biocultural heritage is given in the context of conservation and sustainable landscape management.  This article brings an interesting perspective to  addressing the challenges facing landscapes in Africa and acknowledges the importance of more traditional land use pursuits in preserving landscape diversity and having postive environmental outcomes.  Whilst important to this region the theories are equally relevant to other landscapes in the world. 

I felt the paper was acceptable for publication as is and so have no suggested changes.  My only criticism is that the authors could have made the article more broadly relevant internationally and demonstrated a stronger case for their theories by drawing on a wider base of international literature and emphasising the similarities to other landscape scenarios.

Author Response

Thank you for the good comments we did consider and worked from an earlier manuscript that had a global comparison in. But we found that it distracted the reader to much from the geographical scope dealt with here, eg Subsaharan Africa. Therefore we decided to keep the geopgraphical focus strict even if comparison lies in the references and the concept itself.

Reviewer 3 Report

This application of the concept of Bio-cultural Heritage to the sub-Saharan Africa area is very interesting and the potential implications in the field of conservation are important. There are many similarities with the approach of historical ecology and environmental archeology, even if the authors do not explicitly mention it (but in the bibliography). In this sense I would suggest referring also to Grove Rackham's book The Nature of Mediterranean Europe. An Ecological History, Yale University Press, New Haven 2001.

I appreciate the fact that the authors use the categories of "local practice" and "local practitioner" because they are applicable to individuals, groups, societies, communities, etc. in specific places and times. And also the category of "environmental resources" rather than "natural resources", which is perfectly in line with the overcoming of the nature-culture dichotomy.

I have just a few requests for clarification on the three levels in which the concept of Biocultural Heritage is declined.

1. Clarify better the definitions (also in fig.1): these three categories of memories (Ecosystem memories, Landscape memories, Place based memories) are practices (shifting agriculture, grazing, fire management, etc.) or the effects of these practices (black earth in the soils, grasslands with a specific composition, glades, wood pastures, micro-habitat, etc.) or - as it seems - both?

2. The distinction between Ecosystem memories and Landscape memories seems to me to be sustainable not so much in terms of scale (spatial and temporal) as in the sense that in the first case it is a "biological/ecological" memory (in the soil or in the vegetation cover) while in the second it is more a "land form/earth work" memory (irrigation channels, terraces ..).

3. I find it a bit difficult to understand the definition of Place based memories as there is only one example, that is the practice of protecting the "sacred areas", often coinciding with the ancient burial areas of the community or old settlement sites that in some cases are then become old grow forests. Even in this case, I do not think it is a problem of scale, but, again, of kind of memory (in this case more relate to spirituality than to agriculture?). Perhaps it should be better explained why it could not fit into the Ecosystem memories as it is a practice that has effects on the ecosystem (forest) itself... 

41. [..something is missing

60. Cevasco, R .; Moreno, D. and Hearn ... ..

162. delete "have been" (is repeated)

334. I suggest to add: Krzywinski K., O'Connell M., Küster H.J. (eds), 2009, Cultural Landscapes of Europe. Fields of Demeter, Haunts of Pan, Bremen: Aschenbeck media, ISBN: 978-39-416-2470-2

Author Response

I would suggest referring also to Grove Rackham's book The Nature of Mediterranean Europe. An Ecological History, Yale University Press, New Haven 2001.

RESPONSE: NOW ADDED

1. Clarify better the definitions (also in fig.1): these three categories of memories (Ecosystem memories, Landscape memories, Place based memories) are practices (shifting agriculture, grazing, fire management, etc.) or the effects of these practices (black earth in the soils, grasslands with a specific composition, glades, wood pastures, micro-habitat, etc.) or - as it seems - both?

RESPONSE: A CLARIFICATION HAS NOW BEEN ADDED

2. The distinction between Ecosystem memories and Landscape memories seems to me to be sustainable not so much in terms of scale (spatial and temporal) as in the sense that in the first case it is a "biological/ecological" memory (in the soil or in the vegetation cover) while in the second it is more a "land form/earth work" memory (irrigation channels, terraces ..).

RESPONSE: WE HAVE CLARIFIED THAT WE ARE DECSCRIBING BOTH PRACTICE AND OUTCOME, SO I THINK THIS IS NOW MORE CLEAR

3. I find it a bit difficult to understand the definition of Place based memories as there is only one example, that is the practice of protecting the "sacred areas", often coinciding with the ancient burial areas of the community or old settlement sites that in some cases are then become old grow forests. Even in this case, I do not think it is a problem of scale, but, again, of kind of memory (in this case more relate to spirituality than to agriculture?). Perhaps it should be better explained why it could not fit into the Ecosystem memories as it is a practice that has effects on the ecosystem (forest) itself... 

RESPONSE: WE NOW ADDED AN EXPLANATION CLARIFYING THAT THEY CAN ALSO RESULT IN LANDSCAPE SCALE/ECOSYSTEM MEMORIES WITH AN EXPLANATION

 41. [..] something is missing

RESPONSE: NOW EDITED

60. Cevasco, R .; Moreno, D. and Hearn ... ..

RESPONSE: NOW ADDED

162. delete "have been" (is repeated)

RESPONSE: NOW CHANGED

334. I suggest to add: Krzywinski K., O'Connell M., Küster H.J. (eds), 2009, Cultural Landscapes of Europe. Fields of Demeter, Haunts of Pan, Bremen: Aschenbeck media, ISBN: 978-39-416-2470-2

RESPONSE: NOW ADDED